# A Combined Gas and Water Permeances Method for Revealing the Deposition Morphology of GO Grafting on Ceramic Membranes

**DOI:** 10.3390/membranes13070627

**Published:** 2023-06-28

**Authors:** Evdokia Galata, Charitomeni M. Veziri, George V. Theodorakopoulos, George Em. Romanos, Evangelia A. Pavlatou

**Affiliations:** 1Laboratory of General Chemistry, School of Chemical Engineering, National Technical University of Athens, Zografou Campus, 9, Iroon Polytechniou Str., Zografou, 15780 Athens, Greece; galata@mail.ntua.gr; 2Institute of Nanoscience and Nanotechnology, N.C.S.R. “Demokritos”, Ag. Paraskevi, 15310 Athens, Greece; c.vezyri@inn.demokritos.gr

**Keywords:** GO/composite membranes, ceramic membranes, crosslinkers, deposition morphology, chemical attachment, gas permeance, water permeance

## Abstract

The adhesion enhancement of a graphene oxide (GO) layer on porous ceramic substrates is a crucial step towards developing a high-performance membrane for many applications. In this work, we have achieved the chemical anchoring of GO layers on custom-made macroporous disks, fabricated in the lab by pressing α-Al_2_O_3_ powder. To this end, three different linkers, polydopamine (PDA), 3-Glycidoxypropyltrimethoxysilane (GPTMS) and (3-Aminopropyl) triethoxysilane (APTMS), were elaborated for their capacity to tightly bind the GO laminate on the ceramic membrane surface. The same procedure was replicated on cylindrical porous commercial ZrO_2_ substrates because of their potentiality for applications on a large scale. The gas permeance properties of the membranes were studied using helium at 25 °C as a probe molecule and further scrutinized in conjunction with water permeance results. Measurements with helium at 25 °C were chosen to avoid gas adsorption and surface diffusion mechanisms. This approach allowed us to draw conclusions on the deposition morphology of the GO sheets on the ceramic support, the mode of chemical bonding with the linker and the stability of the deposited GO laminate. Specifically, considering that He permeance is mostly affected by the pore structural characteristics, an estimation was initially made of the relative change in the pore size of the developed membranes compared to the bare substrate. This was achieved by interpreting the results via the Knudsen equation, which describes the gas permeance as being analogous to the third power of the pore radius. Subsequently, the calculated relative change in the pore size was inserted into the Hagen–Poiseuille equation to predict the respective water permeance ratio of the GO membranes to the bare substrate. The reason that the experimental water permeance values may deviate from the predicted ones is related to the different surface chemistry, i.e., the hydrophilicity or hydrophobicity that the composite membranes acquire after the chemical modification. Various characterization techniques were applied to study the morphological and physicochemical properties of the materials, like FESEM, XRD, DLS and Contact Angle.

## 1. Introduction

Graphene oxide (GO) holds great interest within the scientific community, due to the functional groups that are part of its structure. In the basal plane of the GO layers there are carbonyl and epoxy groups, whereas oxygen-containing functional groups, such as hydroxyl and carboxyl groups, emanate from the edges of GO [1,2]. These groups increase the ability for additional modification of the GO structure, opening the path for a multitude of applications. This is despite the functional groups possessed by GO being hydrophilic and negatively charged, making its exfoliation in wet environment easy. This constitutes a major drawback of GO membranes for liquid phase applications. In this case, for enhancing the stability and raising the lifespan of the membrane in a wet state, crosslinking with a substrate offers a viable solution.

Ceramic substrates are the most appropriate choice due to their robustness and ability to operate in aggressive environments. Hence, the stability of the GO and its exceptional mechanical properties combined with the durability of the ceramic substrate make an ideal combination that promises membranes of extended lifetimes [3]. The tight anchoring of the GO laminate on the ceramic substrate is the sole prerequisite towards this target. Consequently, the right selection of linkers is a crucial factor for successfully functionalizing the substrate and developing a robust membrane. In this study, the choice of linkers was based on the demand for the existence of two different functional groups, one which is appropriate for grafting on the ceramic substrate and a second one for anchoring and firmly holding the GO nanosheets. Notwithstanding the importance of such a perspective, there are only a few studies that have dealt with the development of these types of chemically modified membranes and the in-depth examination of their stability in harsh environmental conditions. The complexity of the composite membrane structure constitutes the main reason for the shortage of studies and reports on GO membranes for application in the liquid phase. For instance, Xu et al. [4] successfully functionalized Al_2_O_3_ porous substrates by using polydopamine (PDA) as a covalent linker, and GO nanosheets were further attached onto the support surface for seawater desalination. The water flow through the composite membrane was exceptionally high (48.4 kg m^−2^ h^−1^ at 90 °C) and the ion rejections achieved were over 99.7%. The fact that the water flow and ion rejection performance were steady over a period of fourteen days led researchers to the conclusion that the developed membranes were characterized by extended stability. Another study elaborated the stability of GO-PDA/O=CS/ceramic membranes in water applications and properties, such as the water permeance and dye rejection being monitored for twenty-five days, with very promising results. In this study, PDA was found to be the most effective linker as compared with EDA and PPD, having membranes with remarkable anti-swelling properties in water [5].

In addition, Karim et al. [6] developed a reduced graphene oxide membrane on ceramic pozzolan support using a chemical grafting/spin-coating method. The linker employed as a molecular bridge between the ceramic support and rGO was 3 Glycidoxypropyltrimethoxysilane (GPTMS). The as-derived composite membranes were used for soluble dye removal applications, exhibiting rejection performances of 94, 93 and 97% for bromothymol blue, methyl orange and murexide, respectively. Lou et al. [7] investigated the pervaporation performance of GO/ceramic composite membranes prepared with GLYMO as a linker, targeting to separate water from ethanol/water mixtures. These membranes exhibited a total flux of 461.86 g/(m^2^ h) and a water concentration enhancement from 5 wt.% to 39.92 wt.% at 40 °C.

Another linker, which was used for the attachment of graphene oxide quantum dots (GOQDs) on ceramic substrates towards the development of membranes with antifouling potential, was the (3-aminopropyl) triethoxysilane (APTMS). Water permeability experiments were conducted using the dead-end filtration method at a TMP of 0.1 MPa and showed the enhancement of the pure water flux (~30%), and reduced membrane resistance (~15%) compared to the pristine ceramic membranes. The membranes were also endowed with exceptional resistance to organic fouling, which was validated with the use of humic acid solutions [8].

In our recent work [9], we partially filled some of the knowledge gaps concerning the morphological and pore structural characteristics of chemically modified composite ceramic/GO membranes and how these characteristics are related to the type of the organic linker, the number of GO layers composing the GO laminate and the eventual gas separation performance. This was achieved by designing and implementing a systematic experimental campaign based on the measurements of the permeance of several gas molecules at various temperatures and transmembrane pressures. The motivation for the previous study was the lack of connection between the so-far-reported good performance of GO membranes in several liquid phase applications and the deposition morphology of GO on the chemically modified ceramic support, in conjunction with their pore structural features and the type of functional groups that interact with each other or remain free after chemical attachment. Moreover, it was showcased that several conclusions relative to the manner of linkers’ attachments to the ceramic substrate and the GO nanosheets, as confirmed by a multitude of spectroscopic techniques, were also validated by the gas permeance studies. This was a highly important outcome, since spectroscopic techniques suffer from their local character, which makes it difficult to obtain statistically reliable results without spending a vast amount of analytical time in scanning the entire membrane surface.

In this work, we propose a combined method based on the simultaneous interpretation of gas and water permeance results for revealing the conformation of GO’s anchoring on GO/ceramic composite membranes, developed via the use of several organic linkers. Helium at 25 °C was the probe gas for the permeance measurements that were performed with the closed volume (dead-end) method. The probe gas was selected with the purpose of limiting adsorption and the concomitant contribution of surface diffusion on the overall permeance mechanism. Thence, the results obtained for the GO/ceramic composite membranes were compared with those of the bare ceramic substrate using the Knudsen expression of permeability ($P^{g}$ (m^2^·s^−1^)), as follows [10]:(1)Pg=JlUc∆C=8δ3τk1ε2Ac2RTπM12
where *J* (mol·s^−1^) is the gas molar flux through the membrane, *U_c_* (m^2^) is the surface of the membrane, l (m) is the thickness of the membrane, Δ*C* (mol·m^−3^) is the gas concentration difference across the membrane, *ε* (-) is the porosity (pore volume per total volume (*V*) of the membrane), $A_{c}$ (m^−1^) is the internal surface of the pores per total volume (*V*) of the membrane, τ (-) is the tortuosity, which is indicative of the total length of the pores over the membrane thickness (*L*/l), δ (-) is a factor that approaches $\frac{2-f}{f}$ for low pressure, with *f* being the fraction of gas molecules striking to the pore walls, and $k_{1}$ (-) is a shape factor, which is equal to 1 for cylindrical pores. For cylindrical pores, $\frac{\varepsilon^{2}}{A_{c}}$ can be written in terms of the pore radius, and then Equation (1) is transformed to:(2)Pg=43τNπr3LV2RTπM12
with *N* (-) corresponding to the number of pores and *L* (m) to the real length of the pores. Furthermore, Equation (2) is transformed to the expression of the gas permeance (${Pe}^{g}$ (m·s^−1^)) by dividing the permeability by the thickness of the membrane.
(3)Peg=43τNπr3V2RTπM12

Equation (3) shows that the Knudsen permeance was directly analogous to the third power of the pore radius. It is noted that Knudsen diffusion was considered to be the sole mechanism contributing to the gas flux through the membranes, since all the experiments were performed at low transmembrane pressures (up to 300 mbar). As such, with an average pressure of 150 mbar across the membrane’s sides and at 25 °C, the mean free path *λ* of He was *λ* = 0.9 μm. All the examined membranes had smaller pore sizes than the bare α-Al_2_O_3_ disk, which, as further explained in the discussion of the results, was defined considering the particle size distribution of the α-Al_2_O_3_ powder and was 0.067 μm. Therefore, the pore sizes of all membranes were much smaller than the mean free path of He (<0.1 · *λ*) and as a consequence, mass transfer mechanisms, such as slip flow and viscous flow, were excluded.

Since the gas permeance results confirmed that the main gas diffusion mechanism was Knudsen, the ratio of the pore radius of the blank substrate (*r_s_*) to the pore radius of the modified membrane (*r_m_*) can be calculated from the following equation:(4)$$\frac{{Pe}_{s}^{g}}{{Pe}_{m}^{g}}={\left(\frac{r_{s}}{r_{m}}\right)}^{3}$$

Next, the Hagen–Poiseuille equation was involved to describe the pressure-driven water permeability $P^{l}$ (m^2^·s^−1^) through the membranes. In Equation (5), $\hat{P}$ (Pa) is the average of the pressure at both sides of the membrane and η (Pa·s) is the viscosity of the liquid phase. The shape factor $k_{0}$ (-) takes the value of 2 for cylindrical pores and the factor $\frac{\varepsilon^{3}}{{A_{c}}^{2}}$ is written in terms of the pore radius:(5)$$P^{l}=\frac{\hat{P}}{\tau {\eta k}_{0}}\frac{\varepsilon^{3}}{{A_{c}}^{2}}=\frac{\hat{P}}{8\tau \eta }\frac{Ν\pi r^{4}L}{V}$$

Water permeance, ${Pe}^{l},$ can be derived by dividing the permeability by the membrane thickness. Hence, Equation (5) is transformed to:(6)$${Pe}^{l}=\frac{P^{l}}{l}=\frac{\hat{P}}{8\tau \eta }\frac{Ν\pi r^{4}}{V}$$

Thence, the water permeance of the blank substrate ${Pe}_{s}^{l}$ relative to the permeance of the modified membrane ${Pe}_{m}^{l}$ is:(7)$$\frac{{Pe}_{s}^{l}}{{Pe}_{m}^{l}}={\left(\frac{r_{s}}{r_{m}}\right)}^{4}$$

Knowing the $\frac{r_{s}}{r_{m}}$ ratio from Equation (4) and introducing it in Equation (7), it is possible to predict the water permeance ratio of the bare substrate to the modified membrane. Convergence between the prediction and experimental results implies that only the pore structural features affect the water passage through the modified membranes. However, any positive or negative deviations can provide information on the hydrophilicity or hydrophobicity that the composite membranes acquire after chemical modification. Hence, it is possible to draw conclusions about the manner of chemical attachment of the GO nanosheets with the linker, especially regarding which groups of the GO’s surface participate in the chemical anchoring with the organic linker, and which remain intact and are responsible for the hydrophilic character of the composite membranes. Further to this, by extending the permeance measurement period it was possible to elaborate the stability of the developed membranes in contact with the water stream. The conclusions drawn from the simultaneous interpretation of the gas and water permeance results were further confirmed with other techniques, such as contact angle and zeta potential measurements.

## 2. Materials and Methods

The materials (3-aminopropyl)triethoxysilane 99% (Sigma-Aldrich Chemicals, St. Louis, MO, USA), graphene oxide (GO, Abalonyx AS, Forskningsveien, Oslo, Norway), dopamine hydrochloride (Sigma-Aldrich chemicals, St. Louis, MO, USA), tris-HCl (Sigma-Aldrich Chemicals, St. Louis, MO, USA), (3-glycidyloxypropyl), trimethoxysilane ≥ 98% (Fluorochem Ltd., Hadfield, UK), ethylenediamine (EDA) 99% (Alfa Aesar, Haverhill, MA, USA), titanium (IV) butoxide 99% (Acros Organics, Geel, Belgium), and α-alumina powders (Baikalox, CR-6, Baikowski, La Balme-de-Sillingy, France) were employed.

### 2.1. Preparation of a-Alumina Disks

Lab-made α-alumina discs, 2 mm thick and 22 mm in diameter, were used as supports. The discs were fabricated by pressing commercial α-alumina powders (Baikalox, CR-6) in a custom-made mold with the aid of a hydraulic press (Carver, Inc., Wabash, IN, USA) and sintering at 800 °C for 30 h and further at 1180 °C for 2 h. One side of the disc was polished with SiC sandpaper (Buehler, grit size 600) [11].

#### 2.1.1. Preparation of Composite Membranes with PDA as Linker

A cleaning procedure for Al_2_O_3_ disks was performed by holding them in boiling H_2_O_2_ 30% for 10 min. Hydroxylation of the ceramic surface was achieved with thermal treatment in NaOH (pH = 9.5) at 70 °C for 15 min. Dopamine (2 mg/mL) was dissolved in 10 mM Tris-HCl (pH = 8.5) at room temperature for 20 h, leading to the polymerization of dopamine to polydopamine (PDA) on the surface of Al_2_O_3_. Modification of Al_2_O_3_-PDA with graphene oxide (GO) was achieved with one facile step of dip-coating of composite membrane in GO (1 mg/mL) dispersion [4,12]. The preparation method of composite Al_2_O_3_ PDA-GO membrane is illustrated at Figure 1.

Modification of Al_2_O_3_-PDA with reduced graphene oxide (rGO)–TiO_2_ was held with the preparation of GO-TiO_2_ nanoparticles via the sol-gel method, as depicted in Figure 2. Specifically, 0.1 mg/mL GO/ethanol mixture was sonicated for 90 min. After 20 min of vigorous stirring, 0.4 mL NH_3_ was added. The stirring was continued for another 30 min and 3 mL of tetrabutyl titanate (TBOT) was added. The sol-gel process was kept for 24 h. After that, washing with ethanol 3 times followed. The PDA-treated ceramic support was then dip-coated into a 1 mg/mL GOT/H_2_O solution. A thermal treatment of the ceramic substrate with TiO_2_/GO sheets under Ar atmosphere at 500 °C resulted in the composite membrane. During thermal treatment, the amorphous TiO_2_ nanoparticles were crystallized into uniform anatase nanoparticles, accompanied by the reduction of GO sheets, leading to the formation of TiO_2_ nanocrystals/rGO sheets [13].

#### 2.1.2. Preparation of Composite Membranes with APTMS as Linker

Al_2_O_3_ disks were boiled in H_2_O_2_ 30% for 10 min and then dried at 150 °C for 2 h. After, the ceramic membranes were immersed in ethanol for 10 min, followed by the addition of 0.5 mL of APTMS in 300 mL of ethanol at 25 °C for 90 min under vigorous stirring. Then, the membranes were again thoroughly cleaned with copious amounts of ethanol to remove the unreacted APTMS and dried at 60 °C. The chemical grafting of GO on the surface of the modified membrane was conducted via dip coating method in a GO/H_2_O dispersion (1 mg/mL) and heated at 100 °C for 2 h to allow the linkage between the carboxylic group of GO and amine group of APTMS [8]. The total experimental procedure is depicted in Figure 3.

#### 2.1.3. Preparation of Composite Membranes with GPTMS as Linker

Alpha alumina disks were boiled in H_2_O_2_ 30% for 10 min to introduce hydroxyl groups onto the surface of the ceramic support and then were dried at 150 °C for 2 h. The preparation was followed by the immersion of the ceramic disk in GPTMS/absolute ethanol solution for 30 min at 40 °C. Then, the membrane was heated for 4 h at 110 °C. The modified ceramic was dip-coated into graphene oxide aqueous solution (1 mg/mL) and dried at 50 °C [7]. The preparation method is depicted in Figure 4.

#### 2.1.4. Preparation of Composite Membranes with Vacuum Filtration Method and EDA as Linker between the GO Sheets

Two additional membranes, bearing a thick GO laminate on top of the thin oligo-layered one, were prepared via vacuum filtration of a GO-EDA dispersion through the chemically modified membranes Al_2_O_3_ APTMS-GO and Al_2_O_3_ GPTMS-GO. To achieve enhanced dispersion of the GO nanosheets, a GO 0.1 mg/mL aqueous solution (DI water) was initially subjected to ultra-sonication for 30 min. Then, EDA (5 wt.%) was added into the GO dispersion, followed by ultra-sonication and 16 h stirring at room temperature to end up with a homogeneous solution ready for filtration [14].

At this point, it could be stated that the cylindrical porous commercial ZrO_2_ substrates were prepared with both aforementioned procedures (chemical modification and vacuum filtration), followed the same experimental conditions in order to evaluate the composite membranes’ potentiality in scale-up applications.

### 2.2. Filtering Device

A custom-made device was used to filter an aqueous GO-EDA dispersion through the pores of the composite ceramic membranes. The filtering device was equipped with a vacuum pump, and consisted of a stainless-steel cell, where the membrane was placed, and a trap inserted between the vacuum pump and the permeation side of the membrane cell to collect the filtrate. The GO-EDA dispersion was conveyed into the feed side of the membrane at a constant flow rate using a peristaltic pump and the retentate was recycled back to the stirring vessel that contained the GO-EDA dispersion. Under these conditions of continuous flow and agitation, air bubbles and GO precipitation were avoided and this ensured the creation of a quite homogeneous coating area. Subsequently, the membrane was placed in an oven at 50 °C for drying and promoting the crosslinking between EDA and GO. The drying process was performed gradually to avoid the peeling off of the GO-EDA laminate from the ceramic support.

### 2.3. Gas Permeance Device

Single phase gas permeance experiments were conducted in a home-made stainless steel permeability rig involving the closed volume technique [15]. Briefly, the feed side of the membrane was kept under constant gas pressure, while the permeate side was interfaced with a closed chamber of specific volume, evacuated down to 10^−6^ mbar by a turbo-molecular pump. When the experiment started, gas molecules that were passing through the pores of the membrane caused a pressure increase in the closed permeation chamber. The gas permeance was calculated according to the following equation:(8)$${Pe}^{g}=1.645\times 10^{-6}\frac{\Delta P_{L}}{\Delta t}\frac{V_{L}}{S\;P_{H}\;R\;T_{mem}}\left(\frac{\mathrm{mol}}{m^{2}\sec\mathrm{Pa}}\right)$$
where Δ*P_L_*/Δ*t* (mbar/min) is the rate of pressure increase at the permeate side of the membrane, *S* (cm^2^) is the membrane surface area, *V_L_* (cm^3^) is the volume of the closed chamber at the permeate side of the membrane, which was defined by a procedure of He expansion from a known volume, *P_H_* (mbar) is the constant gas pressure at the feed side of the membrane, *R* is the gas constant (atm L/mol K), and *T_mem_* (K) is the membrane cell temperature. The permeability measurement was continued, on the condition that the feed-side pressure (*P_H_*) remained much larger than that of the permeate side (*P_L_*) [16].

### 2.4. Water Permeability Device

A simple set-up consisting of an HPLC pump, a membrane cell and a metering valve connected to the retentate effluent of the membrane cell was assembled with the purpose of conducting de-ionized (DI) water permeance measurements. The pump was used to convey a water stream of constant flow rate at the feed side of the membrane, while the feed pressure was monitoring. For each flow rate, the metering valve was regulated to keep the feed side pressure below 15 bars. The flow rates of the permeate and retentate effluents were calculated by logging the time required for collecting a specific volume of water. The water permeance ${Pe}^{l}$ (Lm^2^h/bar) was derived according to the following equation:(9)$${Pe}^{l}=\frac{Vl}{\left(A\;\Delta t\;\Delta p\right)}$$
where *V* (L) represents the water volume collected at the permeate side, *A* (m^2^) is the membrane’s effective surface area, and Δ*p* (bar) and Δ*t* (h) are the transmembrane pressure and sampling time, respectively [17].

### 2.5. Characterization Techniques

The structural properties of the chemically modified GO membranes were elaborated via X-ray Diffraction (XRD) analysis (D8 Advance, Bruker, Germany). The measurements were performed at 2-theta angles between 5° and 80° and a scanning rate of 0.03°/min, employing Cu-Kα radiation (λ = 1.5418 Å) at a voltage of 30 kV and a current of 15 mA. The morphology of ceramic modified membranes was examined by using Field Emission SEM (FESEM, JSM-7401F, JEOL, Tokyo, Japan). Each sample was pretreated via gold sputtering prior to the FESEM observation. Zeta potential determination was implemented with a Zeta Sizer nano Series instrument (Malvern Inst., Malvern, UK), which applied a combination of laser Doppler velocimetry and phase analysis light scattering (PALS) in a patented technique called M3-PALS to measure particle electrophoretic mobility. Contact angle measurements were conducted with a contact angle meter (CAM 100, KSV Instruments Ltd., Helsinki, Finland) with the target of gaining insight into the hydrophilicity or hydrophobicity of the composite membranes.

## 3. Results

### 3.1. Structural and Morphological Properties

XRD was used in order to define the d-distance of the GO stacks on the oligo-layered and multi-layered membranes, the latter developed by post filtration of a GO-EDA dispersion through the oligo-layered ones using Bragg’s Law equation [18]:(10)n λ=2 d sinθ
where *n* is an integer, *d* is the distance and *λ* is the wavelength. The average crystallite size of the as-produced composite membranes was calculated by using Scherrer’s equation [19]:(11)$$D=\frac{0.89\;\lambda }{\beta \;cos\theta }$$
where *D* is the average crystallite size, 0.89 is the Scherrer’s constant, *λ* is the X-ray wavelength, *θ* is the diffraction angle, and *β* is the FWHM (full-width-half-maximum). Figure 1 depicts the XRD spectra of all composite Al_2_O_3_/GO membranes, developed with different crosslinkers. The average crystallite size was calculated from the main peak of graphene oxide at 2*θ* = 10°. The respective size of all the thin films was estimated at 9.01 nm, 4.19 nm, 4.06 nm, 5.84 and 4.67 nm for Al_2_O_3_ GPTMS-GO, Al_2_O_3_ APTMS-GO-F, Al_2_O_3_ PDA-GO, Al_2_O_3_ GPTMS-GO-F and Al_2_O_3_ APTMS-GO composite membranes, respectively. The other sharp peaks at the angles of 25° to 80° were representative of α-Al_2_O_3_ [20]. The broad peaks at 24° corresponded to the multi-layered membranes prepared using the post filtration method, as this peak indicates the successful anchoring of GO nanosheets with EDA [21].

It is noticeable that the determination of the d-distance is crucial in supporting the interpretation of gas and water permeance measurements and enabling the clarification of the pore structural properties of the developed membranes. Particularly, for the oligo-layered Al_2_O_3_ GPTMS-GO, Al_2_O_3_ APTMS-GO and Al_2_O_3_ PDA-GO membranes, the d-distance was determined to be 8.45, 8.5 and 9.57 Å, respectively, and for the multi-layered Al_2_O_3_ GPTMS-GO-F and Al_2_O_3_ APTMS-GO-F, d-spacing was 10.39 Å and 12.49 Å, respectively. The main difference between oligo- and multi-layered membranes was that the latter bear a thick GO laminate on top of a thin oligo-layered one. Owing to its much higher thickness, most of the information derived by XRD corresponded to the thick multi-layered laminate. Within the structure of the thick laminate, GO nanosheets are held together with EDA, a linker with two anchoring groups and small molecular size. The latter is the reason behind the small d-distance concluded for membranes Al_2_O_3_ GPTMS-GO-F and Al_2_O_3_ APTMS-GO-F (10.3–12.5 Å). However, in the oligo-layered membranes, the d-distance was much lower (8.5–9.5 Å). In this case, the three different linkers (PDA, APTMS and GPTMS) were used as molecular bridges between the ceramic substrate and the first sheets of GO that were directly attached on the ceramic surface. As such, the information obtained using the XRD technique was related to the d-distance of the subsequent few deposited GO layers. The GO nanosheets composing these layers interacted with each other with physical Van der Waals bonds, and this explains the lower d-distance calculated for membranes Al_2_O_3_ GPTMS-GO, Al_2_O_3_ APTMS-GO and Al_2_O_3_ PDA-GO. The charge of the Al_2_O_3_ substrate and the concomitant attractive or repulsive forces that it may exert on the deposited GO nanosheets can also affect the d-distance of the oligo-layered membranes. This is possibly the reason behind the higher d-distance of the composite membranes that were developed with the PDA linker. PDA sprawls on the ceramic substrate via self-polymerization cyclization. As a result of the bulkier PDA polymer structure compared to the short-chain linkers APTMS and GPTMS, GO sheets were deposited at a longer distance from the surface, and consequently, the interaction of the Al_2_O_3_ surface with the GO sheets became weaker. Explanatively, PDA features reactive alicyclic amine groups and is endowed with a molecular configuration of high steric hindrance that bears less electrostatic repulsion between the similarly charged surfaces of Al_2_O_3_ and the first GO layer. Thus, the stronger binding of the first GO layer on the surface causes the Van der Waals interactions with the succeeding layers to become less significant, leading to much higher d-spacing (9.6 Å) compared to the other two linkers.

FESEM images obtained from the cross sections of the composite GO–ceramic membranes developed on the α-Al_2_O_3_ disks are presented in Figure 2. Independently of the organic linker, all the membranes held a continuous GO layer on top of their surface. Figure 2a,b,d depict the deposition morphology of the GO on the oligo-layer membranes, with the thickness of the GO laminates being estimated at ~100 nm, 38 nm and 200 nm for Al_2_O_3_ APTMS-GO, Al_2_O_3_ GPTMS-GO and Al_2_O_3_ PDA-GO, respectively. Figure 2c depicts the morphology of the multi-layer Al_2_O_3_ GPTMS GO-F, where a large difference in the number of GO layers as compared to the oligo-layered membranes is discernible. The thickness of the formed multi-layered GO laminate can be estimated at ~4700 nm.

Figure 3 illustrates FESEM cross-sections of the composite GO–ceramic membranes that were developed on the commercially available tubular ZrO_2_ substrates with the use of different intermediate linkers. Comparing slightly tilted images (i.e., Figure 2d and Figure 3d), it can be observed that the GO laminates formed on the ZrO_2_ substrates had a smoother surface compared to those deposited on the α-Al_2_O_3_ substrates. This can be attributed to the smaller pore size and higher smoothness of the ZrO_2_ layer as compared to the rough surface of the Al_2_O_3_ disk, consisting of coarse particles, which are closely packed together. Particularly, Figure 3b,d depict membranes ZrO_2_ PDA-GO and ZrO_2_ GPTMS-GO and Figure 3a,c show the morphology of the multi-layered membranes ZrO_2_ APTMS GO-F and ZrO_2_ GPTMS GO-F, respectively. The existence of GO nanosheets on the top surface of the substrate is clearer in the case of the oligo-layered membrane developed with the PDA linker (Figure 3b,d), whereas, as was also the case with the Al_2_O_3_ substrates, the multi-layered membranes developed on the ZrO_2_ substrates bear a quite thick GO laminate. As such, the thickness of the deposited laminates was estimated at ~350 nm for ZrO_2_ APTMS GO-F, 75 nm for ZrO_2_ GPTMS-GO, 55 nm for ZrO_2_ PDA-GO and 400 nm for ZrO_2_ GPTMS GO-F. Despite their small thickness, the structure of the GO laminates seems to be free of cracks and pinholes, as depicted in Figure 3d,e. Figure 3f demonstrates domains of the cross-section and the top surface of the deposited PDA-rGOT laminate simultaneously. It can be seen that the composite membrane ZrO_2_ PDA-rGOT exhibited small sheets of rGO on its surface, something that was not attainable in the case of the Al_2_O_3_ PDA-rGOT membrane, due to the decomposition of rGO.

### 3.2. Surface Properties

To determine the zeta potential (ζ) and gain insights relative to the surface charge of the oligo-layered GO/α-Al_2_O_3_ membranes, the same α-alumina powder, which was used to develop the α-Al_2_O_3_ substrates, was also subjected to a procedure of grafting with the organic linkers followed by the chemical attachment of GO. In this way, the chemical composition of the membranes’ surface was simulated with a material in the powder form, which was further used in the zeta potential measurements. Hence, we had the possibility of defining the negative charge that had been created by the ionization of the various GO’s functional groups. The magnitude of the zeta potential gives an indication of the potential stability of the colloidal systems occurring in the dispersions of the GO/α-Al_2_O_3_ particles. If all the particles in suspension have a large negative or positive zeta potential, then they will tend to repel each other and there will be no tendency for flocculation, sedimentation, or coagulation. As such, the general dividing line between stable and unstable suspensions is generally taken at either +30 or −30 mV. Our study on the GO-modified α-Al_2_O_3_ powders indicated that the zeta potential was pH-dependent and that they are more stable in a basic environment [22]. The charges of the Al_2_O_3_ PDA-GO, Al_2_O_3_ APTMS-GO and Al_2_O_3_ GPTMS-GO were measured at three different pHs, 3.5, 6.5 and 9, as shown in Figure 4.

The negative charge of the membranes is an important property for applications, such as the adsorption of positively charged pollutants [23], the rejection of negatively charged dyes via the Donnan effect [24], for sensing applications [25], etc. According to Konkena et al. [26], the zeta potential of a GO dispersion is also pH dependent, and the highest ζ value of GO was −54.3 mV at pH 10.3. The composite powders in our study exhibited the highest ζ value of −48 mV at pH 9 and were measured for the sample Al_2_O_3_ APTMS-GO. Li et al. [27] reported that the GO sheets exhibited a highly negative charge when dispersed in water, which came as a result of the ionization of the carboxylic acid and phenolic hydroxyl groups that are known to exist on the GO sheets. Their study concluded that the formation of stable GO dispersions should be attributed to electrostatic repulsion, rather than just to the hydrophilicity of GO. Based on the abovementioned data, an explanation can be given for the fact that amongst the three prepared samples, the Al_2_O_3_ APTMS-GO was the one with the most negatively charged surface. Hence, the APTMS linker attaches to α-Al_2_O_3_ through condensation reactions between its alkoxy groups and the hydroxyl groups available on the α-Al_2_O_3_ surface. The amine groups of APTMS remain intact, and being highly nucleophilic they attack the epoxide rings on GO. Thus, the carboxylic acid and phenolic hydroxyl groups on the surface of GO remain unreacted and through their ionization they are responsible for the highly negative charge of Al_2_O_3_ APTMS-GO. It is also important to note that the zeta-potential results were in accordance with the conclusions drawn in our previous study [7], relative to the manner of the linker’s grafting on the α-Al_2_O_3_ surface. As a matter of fact, the zeta potential results showed that in the Al_2_O_3_ GPTMS-GO composite, some of the highly ionizable groups of GO (-OH, -COOH) that were dissociated, causing the strongly negative surface charge, must have been consumed during the interlocking with the organic linker. For instance, when acting as a molecular bridge between GO and alumina, GPTMS follows an anchoring mechanism, which differs substantially from that of APTMS. From there, GPTMS’s grafting took place through condensation reactions of one or two of its methoxy groups with the aluminol groups on Al_2_O_3_, along with the cleavage of the C–O–C linkage and bonding with the Al_2_O_3_ surface at a second aluminol site. With this grafting conformation, the further anchoring of GO could only take place on unreacted methoxy groups of GPTMS, which were attacked by the hydroxyl groups of the GO surface. On this basis, -OH groups (phenolic) on the surface of GO were consumed in condensation reactions, which explains the less negative zeta potential values of Al_2_O_3_ GPTMS-GO as compared to Al_2_O_3_ APTMS-GO. Furthermore, it must be noticed that according to Alheshibri et al. [28] the isoelectric point for Al_2_O_3_ is at pH 8. The acidic nature of the nanoparticles causes a significant amount of H^+^ ions to be present when the pH is lower than 7, and when the pH is more than 7, due to the basic nature of the nanoparticles, a significant amount of OH^-^ ions are present. As such, the effect of the α-Al_2_O_3_ substrate on the surface charge of the composites may be significant for these composite membranes; it seems that, due to the significant thickness of the GO laminates, Al_2_O_3_ effects are masked by the strongly ionizable character of the GO’s oxygenated groups.

Contact angle measurements were carried out on three different sections of the composite GO/ceramic membranes prepared in this work. The average values were presented in Table 1, and further illustrated in the histogram of Figure 5, together with images of the water drop in contact with the surface of the respective samples.

As observed in Figure 5, the most hydrophilic amongst the membranes (of those that could be measured) was the Al_2_O_3_ APTMS-GO-F, having a contact angle of 63°, whereas the least hydrophilic one was Al_2_O_3_ GPTMS-GO-F, with a contact angle of 80°. Considering the results of the zeta potential measurements and the way that the APTMS and EDA linkers anchor the GO nanosheets, i.e., through the nucleophilic attack of their amine groups on the epoxide rings on GO, the enhanced hydrophilicity was an expectable asset for membrane Al_2_O_3_ APTMS-GO-F. This is because after their chemical anchoring with the linkers, the GO nanosheets that compose both the oligo-layered and multi-layered laminates preserve their hydroxyl and carboxyl groups, which are responsible for the highly hydrophilic character of GO. However, for the same reasons, Al_2_O_3_ GPTMS-GO-F membrane should also be highly hydrophilic. A very thick (4700 nm) multi-layered GO laminate constituted the top layer of this membrane, and within this laminate the GO nanosheets were held together by EDA. We should consider that apart from the hydrophilicity or hydrophobicity of the material that composes the membrane, there are many other factors that affect the contact angle measurement, such as the surface roughness, the pore structural characteristics (porosity, pore size, tortuosity), the homogeneity in the distribution of functional groups, and the possible existence of impurities [29]. Therefore, the deviation of the contact angle measurement results from the expected trend will be further discussed in relation to the gas and water permeance results.

### 3.3. Analysis of He and H_2_O Permeability

#### 3.3.1. Comparison of the Bare Substrates

The first issue to discuss concerning the gas and water permeance results relates to the significant differences observed between the bare substrates (α-Al_2_O_3_ disk and tubular ZrO_2_ membrane). The α-Al_2_O_3_ disk was formed from a milled powder, which according to the provider consisted of a 100% α-Al_2_O_3_ crystalline phase and was characterized by a bimodal particle size distribution (PSD), with the larger population frequencies of the particles centered around the sizes of 0.2 and 1 μm. The tapped density of the powder was 0.8 g/mL and the *BET* surface area was 6 m^2^/g. Based on the tapped density of the powder and the solid density of α-Al_2_O_3_ (3.95 g/mL), the pore volume (*PV*) was calculated to be 0.99 mL/g. The pore volume and *BET* surface could give a first insight into the average size of the pores, assuming cylindrical geometry, i.e., $r_{Al}\;\left(nm\right)=\frac{2\cdot PV\cdot 10^{3}}{BET}$, which gives a pore radius of 330 nm. Considering the rule of ~1/3 for the size of the voids between randomly packed particles relative to the size of the particles [30], this value seems to be quite large, almost double that expected for particles of 1 μm. This shows that information on the average pore radius should be mostly derived by considering the bimodal PSD and the rule of 1/3. Thus, by following an averaging approach of the population frequencies of the α-Al_2_O_3_ particles with the two different sizes, an average pore radius of 100 nm can be derived for the bare α-Al_2_O_3_ substrate. Given that the pore radius of the ZrO_2_ substrate is 1.5 nm, we have (from Equation (4)) $\frac{{Pe}_{Al}^{g}}{{Pe}_{Zr}^{g}}={\left(\frac{r_{Al}}{r_{Zr}}\right)}^{3}\approx 3\times 10^{5}$, which is much larger that the ratio of the experimental He permeances (~3.4, Table 2). Even if we compare with the ratio of the gas permeabilities $\frac{P_{Al}^{g}}{P_{Zr}^{g}}$ (Equation (2)), by taking into account that the thickness of the α-Al_2_O_3_ disk is 2000 times larger than that of the ZrO_2_ membrane (2 mm, compared to 1 μm), the deviation still remains very high. Other pore structural characteristics included in Equation (2), such as the number of pores per volume of the membrane (*N*/*V*) and the tortuosity (*τ*), cannot justify such large differences. Hence, it becomes evident that due to the quite different size of the two particles’ populations, smaller particles of 0.2 μm either fill the free space between skeleton particles of 1 μm or wedge in the skeleton of large particles, making the structure of the α-Al_2_O_3_ disk denser and less porous [31]. As such, revisiting the calculation of the permeabilities’ ratio with the acceptance that the pore size of the α-Al_2_O_3_ disk is totally defined by the interstitial space of the smaller particles, we obtained $\frac{P_{Al}^{g}}{P_{Zr}^{g}}={\left(\frac{r_{Al}}{r_{Zr}}\right)}^{3}\approx 10^{4}$, which was in accordance with the experimentally defined value of ~7 × 10^3^. In fact, the small discrepancy can be attributed to differences in the porosity and tortuosity between the two membranes. However, these cannot be so significant as to affect the consensus of our analysis. Therefore, we concluded that in order to perform a reliable interpretation of the water permeability results, the $\frac{r_{m}}{r_{s}}$ must first be derived from the gas permeability rather than from the gas permeance measurements. In this context, the thickness of the deposited GO laminates must be known.

Next, we expanded the discussion on the water permeability results of the two bare substrates. Introducing the calculated result from the gas permeability experiment $\frac{r_{Al}}{r_{Zr}}$ ratio of ~19 to Equation (5), the expected water permeability ratio $\frac{P_{Al}^{l}}{P_{Zr}^{l}}$ must be of the order of 1.3 × 10^5^. Despite this, the experimental results gave a $\frac{{Pe}_{Al}^{l}}{{Pe}_{Zr}^{l}}$ ratio of 10 (see Figure 6) and, accordingly, a $\frac{P_{Al}^{l}}{P_{Zr}^{l}}$ ratio of 2 × 10^4^, something which was consistent with the much higher hydrophilicity of ZrO_2_ as compared to α-Al_2_O_3_ membranes, an asset which is often reported in the literature [32].

Having concluded that the proposed analytical concept concludes with reasonable results both in terms of the gas permeability ratio and the predicted water permeability ratio, we further advanced with the discussion on the composite GO/ceramic membranes.

#### 3.3.2. Comparison of the GO–Ceramic Composite Membranes

Before starting the discussion on the characteristic properties of each membrane, with the focus being mostly on their surface and pore structural features, it must be noted that the gas permeance values (first column, Table 2) of all GO–ceramic composite membranes were significantly lower than those of the respective substrates.

In some cases, the permeance, which corresponds to the gas flux through the membrane, was attenuated by one or two orders of magnitude relative to the bare substrate, and this signifies that the gas flux is totally controlled by the deposited GO laminate (see Table 3 for the entire set of results). As such, the gas permeability can be calculated by multiplying the permeance with the thickness of the GO laminate, the latter being defined by the cross-section images obtained by SEM. It is also important to note that by following the procedure described in Section 3.3.1 and having available the pore size of the bare substrates, it was possible to make an estimation of the average pore dimension in the formed GO laminates.

To be more consistent, the pore dimension in GO membranes related to the distance between parallel-arranged GO layers that formed inter-layer galleries for gas or liquid transport, or to the size of the gaps between stacks of GO nanosheets or to the size of defects existing on the surface of the GO nanosheets. According to the outcome (Table 2), most of the developed GO laminates could be classified as nanofiltration or loose RO membranes. Therefore, if the membranes developed in this work were also endowed with high hydrophilicity and stability in water, they may constitute excellent candidates for a variety of nanofiltration applications, including the rejection of divalent and possibly monovalent ions and the rejection of organic molecules with MW in the range of 200 Da. Hence, it is of high importance that in this work the elaboration of the hydrophilicity of the deposited laminates relative to that of the bare substrate derives the interpretation of the gas and water permeability results as an outcome of this work.

The first of the two last columns in Table 2 present the expected values of the water permeability ratios between the bare substrate and the composite membranes in case the sole effect on the water flux was a result of the different pore structural properties. As such, higher values of the experimentally derived ratios (last column in Table 2) indicate a lower hydrophilicity of the composite membrane compared to the bare substrate, and vice versa. The results confirm that except for two cases, those of membranes ZrO_2_ GPTMS-GO and Al_2_O_3_ PDA, all the other composite membranes exhibited much higher hydrophilicity than the respective α-Al_2_O_3_ and ZrO_2_ membranes. Next, we attempted to rank all the membranes, including the bare substrates, from the least to the most hydrophilic. As an indicator for this ranking, we adopted the extent of deviation of the experimentally derived water permeability ratios from those predicted (second column, Table 4).

The results in Table 4 are presented relative to the α-Al_2_O_3_ substrate, the hydrophilicity indicator factor of which was set to 1. It can be concluded that there was not any relation between the organic linker and the hydrophilic character of the membranes. It is also reasonable that Al_2_O_3_ PDA was less hydrophilic than the bare α-Al_2_O_3_ substrate, because polydopamine sprawls over the entire surface of the membrane, covering all the hydrophilic aluminol groups, whereas when dopamine undergoes oxidative polymerization and cyclisation to PDA, the nucleophilic nitrogen atom of its primary amine group reacts with one carbon of the catechol ring, forming a five-membered ring with the nitrogen enclosed as a heteroatom. As a result, PDA ends up having only secondary amine groups, which are less hydrophilic than the primary ones. It must also be clarified that despite the membrane Al_2_O_3_ PDA-rGOT showing significant hydrophilicity, it must be excluded from further discussions. The reason is that in this specific development we failed to achieve the formation of a continuous rGOT layer. The results presented in Figure 7 depict the gas permeance of Al_2_O_3_ PDA-rGOT as higher than that of Al_2_O_3_ PDA, implying that the process we followed to deposit and reduce GOT to rGOT damaged the already-formed PDA layer.

Thus, in this membrane, there was not a continuous rGOT layer to control the gas and water flux, but rather small domains of rGOT, which are randomly distributed on the entire substrate surface. Having such conformation, the membrane exerts a very low resistance to the gas and liquid flow through its structure. This fact, in combination with the much higher hydrophilicity of TiO_2_ as compared to Al_2_O_3_, leads to the conclusion of the calculation of an enhanced hydrophilicity factor for Al_2_O_3_ PDA-rGOT. As such, having excluded Al_2_O_3_ PDA-rGOT, what is conclusive from the results of Table 3 is that independent of the organic linker, the multilayered membranes are much more hydrophilic than the oligo-layered ones. The ranking of hydrophilicity as deduced from the conjunctive interpretation of the gas and water permeability results was in accordance with the results obtained from the contact angle measurements (see Figure 5), except for the case of membrane Al_2_O_3_ GPTMS-GO-F. In Section 3.2, we explained the reasons for expecting that this membrane would exhibit similarly strong hydrophilicity as membrane Al_2_O_3_ APTMS-GO-F. Whereas in this work the proposed methodology confirmed the expected trends, the contact angle measurements deviated to a high extent. This is because the contact angle measurements could only provide localized information for the selected spots on the membrane. In some cases (see Table 1), we had deviations of more than 10° between the three spots that were examined on each membrane. On the other hand, the information obtained from the water permeability measurements was representative for the entire membrane and much more inclusive, as it pertained not only to the external surface properties of the thin separation layer (GO laminate, ZrO_2_ layer), but also to the bulk structure of the membrane. Equally, the method was informative for water diffusion mechanisms that arose due to the intrinsic hydrophilicity and the sub-nanometer diffusion channels of the GO laminates that led to the friction-free, ultrafast movement of water molecules through the interlayer galleries [33,34]. In our case, the ZrO_2_ GPTMS-GO-F membrane was endowed with the highest hydrophilicity and smaller interlayer space (see Table 4). It seems that the sub-nanometer interlayer distance readily amplified the slip length of water molecules through the interlayer galleries, facilitating frictionless water molecule transport. As such, owing to its sub-nanometer scale interlayer galleries and high hydrophilicity, ZrO_2_ GPTMS-GO-F can constitute a very effective nanofiltration membrane for a variety of applications, with the prerequisite that it preserves its structure and performance for extended periods on stream.

#### 3.3.3. Stability of the GO–Ceramic Composite Membranes

Apart from generating the results, which were essential to validate the proposed methodology via the use of the Hagen-Poiseuille equation, the water permeability experiments provided additional information on the anti-swelling properties of the deposited GO laminates and the mechanical stability of the entire composite membranes. The mechanical stability was mostly related to the capacity of the ceramic substrate to withstand enhanced transmembrane pressure (TMP), which is a highly important asset in nanofiltration membrane technology, because as the pore size decreases the TMP must increase to maintain a significant stage cut. In Figure 8, we present in detail the results obtained for the membranes developed on the α-Al_2_O_3_ disks using PDA as the linker, focusing mostly on the oligo-layered ones, while in Figure 9 a general overview of the results is provided for all the composite GO/ceramic membranes of flat geometry (α-Al_2_O_3_ disks).

It can be seen (Figure 8) that the oligo-layered GO laminate membranes developed with the use of PDA exhibit lower permeance than the bare Al_2_O_3_ substrate. Nevertheless, it must be noted that the bare substrate was not stable and broke at 120 min due to the enhanced pressure difference that was built as a result of the forced DI water flow through the membrane. The next step of modification was the self-polymerization of PDA on the Al_2_O_3_ surface, which made the Al_2_O_3_ PDA membrane more stable, but reduced the permeance by approximately 1.6 times. This membrane was subsequently functionalized by grafting GO nanosheets on its surface, something that further decreased the permeance of Al_2_O_3_ PDA-GO in relation to the other two. The reason that the permeance decreased cannot be attributed to the induction of hydrophobic characteristics since, as presented in Table 4, Al_2_O_3_ PDA-GO was in the middle of the ranking of all membranes according to their hydrophilicity index. In addition, due to the low permeance of Al_2_O_3_ PDA-GO, we tried to enhance the permeate flux by increasing the TMP at 12 bar, and the membrane was broken. On the other hand, Al_2_O_3_ PDA-rGOT presented almost the same permeance as the substrate (Figure 8), due to the lack of a continuous rGOT laminate that was attributed to the decomposition of PDA at 500 °C, a temperature required to transform amorphous TiO_2_ to the anatase crystalline phase. Regarding the mechanical stability, it should be noticed that the breaking of the membranes at TMPs above 10 bar, something that often took place in the case of the GO/α-Al_2_O_3_ samples, never occurred with the membranes developed on the ZrO_2_ substrates. This implies that tubular geometry is the best choice when trying to tailor the pore size of nanofiltration membranes at the sub-nanometer scale, because ceramic tubes can withstand much higher TMPs compared to flat plates. Reverting to the water permeance results, a common characteristic of the oligo-layered GO/α-Al_2_O_3_ membranes, such as the Al_2_O_3_ PDA-GO and the Al_2_O_3_ GPTMS-GO (Figure 9), is that the water permeance is very low and, in addition, it is subjected to a further sudden decrease during the first minutes on stream. We can attribute this behavior to a rearrangement of the GO sheets within the GO laminate, which brings the GO layers closer under the application of high TMPs. Specifically, in the case of the oligo-layered membranes, all subsequent layers are on top of the first one, which is chemically anchored on the substrate’s surface, and are held together through weak Van der Waals forces. In the absence of linkers, the distance between the layers can become very short, allowing the interlocking of their edges through the interaction of the functional groups nesting at the edges of the GO nanosheets. This significantly impedes the intercalation of water molecules into the interlayer galleries and, in addition, the space of the inter layer galleries can be continuously narrowed as the TMP increases. On the other hand, all multi-layered membranes, especially the ones developed on the ZrO_2_ substrates (Figure 10 and Figure 11), exhibited significantly stable water permeance values, since all GO layers were strongly held together by the EDA linker prepared by the vacuum filtration method. Thus, swelling or shrinking behavior was excluded in this type of membrane.

## 4. Conclusions

In this study, ceramic chemically modified membranes with GO, in two different supports (Al_2_O_3_, ZrO_2_) with three different crosslinkers (GPTMS, APTMS and PDA), were successfully prepared. In order to study their morphological and physicochemical properties, various characterization techniques were employed such as FESEM, XRD, DLS and Water Contact Angle.

All membranes held a continuous GO layer on top of their surface that was free of cracks and pinholes. The thickness of the GO nanosheets for the oligo-layered membranes fluctuated from 38 nm to 200 nm, and for multi-layered ones it was ~4700 nm. Among the composites, the Al_2_O_3_ APTMS-GO exhibited the most negatively charged surface due to the way that GO laminates were chemically anchored on the ceramic surfaces. In particular, the carboxylic acid and phenolic hydroxyl groups on the GO surface remained unreacted, resulting in the highly negative charge of composites via their ionization, whereas in Al_2_O_3_ GPTMS-GO the OH groups on the GO surface were consumed in condensation reactions, leading to a lower negative zeta potential value than the Al_2_O_3_ APTMS-GO.

Furthermore, the extensive combined single gas permeability of Helium at 25 °C and a water permeability study were performed to reveal the deposition morphology of GO on the ceramic support, the mode of chemical bonding with the crosslinker, and the water stability. The results confirmed that except for membranes ZrO_2_ GPTMS-GO and Al_2_O_3_ PDA, all the other composite membranes exhibited much higher hydrophilicity than the respective α-Al_2_O_3_ and ZrO_2_ substrates.

The high hydrophilicity of ZrO_2_ GPTMS-GO-F and its sub-nanometer scale interlayer galleries rendered it an effective nanofiltration membrane for a plethora of applications, with the prerequisite that it preserves its structure and performance for extended periods on stream. Overall, the tubular geometry was the best choice for tailoring the pore size of the nanofiltration membranes at the sub-nanometer scale, as ceramic tubes can withstand much higher TMPs compared to the Al_2_O_3_ flat substrates.

## Data Availability

Not applicable.

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
