# Peer review of "A Combined Gas and Water Permeances Method for Revealing the Deposition Morphology of GO Grafting on Ceramic Membranes"

_membranes, 2023, doi:10.3390/membranes13070627_

Round 1

Reviewer 1 Report

Journal Name: Membranes-MDPI

Title: A combined gas and water permeances method for revealing the deposition morphology of GO grafting on ceramic membranes

In the current research, E.A. Pavlatou et al. did good research on membranes. In order to create high-performance membranes for a variety of uses, this work focuses on improving the adhesion of the graphene oxide (GO) layer on porous ceramic substrates. The GO layers were chemically attached to custom-made macroporous disks made from -Al2O3 powder by the researchers. Polydopamine, 3-Glycidoxypropyltrimethoxysilane, and (3-Aminopropyl) triethoxysilane were the three different linkers they looked into for securing the GO laminate to the ceramic membrane's surface. For the purpose of scalability, the same procedure was carried out on cylindrical, porous commercial ZrO2 substrates. To avoid the effects of gas adsorption and surface diffusion, helium at 25 °C was used to test the membranes' gas permeance properties. The results of water permeance were also looked at. However, due to differences in surface chemistry caused by chemical modification, such as hydrophilicity or hydrophobicity, the experimental water permeance values may differ from the predicted values. To investigate the morphological and physicochemical properties of the materials, a variety of characterization methods, including contact angle measurements, FESEM, XRD, and contact angle, were utilized.

This work is interesting and matches the scope of MDPI Membranes. However, the article needs to be improved further to reach the general audience of MDPI membranes. Based on the above considerations, a minor revision is given. Here are some suggestions.

Major revision

1.      I suggest keeping a graphical abstract

2.      Kindly cite relevant articles from 2023; authors must also cite relevant MDPI sensors articles. (While discussing the recent graphene-related articles published in 2022 (Pre-post redox electron transfer regioselectivity at the alanine modified nano graphene electrode interface, Chemical Physics Letters, 789, 2022, 139295))

3.      The quality of Fig.9 has to be improved

4.      Citation required for Equation 8 and 9

5.       How did the surface chemistry of the composite membranes change after the chemical modification? Were the membranes more hydrophilic or hydrophobic compared to the bare substrate? Discuss contact angle results with few more references

6.      What were the key findings from the characterization techniques used (FESEM, XRD, DLS, Contact Angle)? Did they provide insights into the morphological and physicochemical properties of the materials? Discuss it further in the conclusion

7.      In terms of scalability, how well did the procedure replicate on cylindrical porous commercial ZrO2 substrates? Were there any challenges or notable differences compared to the custom-made macroporous disks?

Reviewer 2 Report

This work studied the fabrication and characterization of GO-grafted ceramic membranes. A combined gas and water permeances method was applied for revealing the morphology of the prepared composite membrane. The manuscript is in general well written and structured. It can be accepted after addressing my following concerns:

1. The d-distance of the GO membrane layer is much smaller than the mean free path of helium molecules. Are there other forms of mass transfer such as slip flow? In the manuscript, the Knudsen diffusion model is used, and the explanation of its rationality needs to be further strengthened.

2. The quality of the cross-sectional FESEM images (Fig. 2, 3) of the composite membrane needs to be improved. In this version, it is difficult to correspond to the text description. In addition, as seen from the images, the membrane layers are peeled off from the support, and there are many particles on the surface.

3. Is the subgraph of the water droplet contact angle in Fig 5 stretched?

4. To obtain the pore dimension by the permeance, the tortuosity must be specified and evaluated.
